# Targeted Therapy of Multiple Sclerosis: A Case for Antigen-Specific Tregs

**DOI:** 10.3390/cells13100797

**Published:** 2024-05-08

**Authors:** Yiya Zhong, Hans J. Stauss

**Affiliations:** Institute of Immunity and Transplantation, Division of Infection and Immunity, University College London, Royal Free Hospital, Rowland Hill Street, London NW3 2PP, UK; yiya.zhong.22@ucl.ac.uk

**Keywords:** autoimmunity, multiple sclerosis, immunotherapy, regulatory T cells, T cell receptor, adoptive T cell therapy

## Abstract

Multiple sclerosis is an autoinflammatory condition that results in damage to myelinated neurons in affected patients. While disease-modifying treatments have been successful in slowing the progression of relapsing–remitting disease, most patients still progress to secondary progressive disease that is largely unresponsive to disease-modifying treatments. Similarly, there is currently no effective treatment for patients with primary progressive MS. Innate and adaptive immune cells in the CNS play a critical role in initiating an autoimmune attack and in maintaining the chronic inflammation that drives disease progression. In this review, we will focus on recent insights into the role of T cells with regulatory function in suppressing the progression of MS, and, more importantly, in promoting the remyelination and repair of MS lesions in the CNS. We will discuss the exciting potential to genetically reprogram regulatory T cells to achieve immune suppression and enhance repair locally at sites of tissue damage, while retaining a fully competent immune system outside the CNS. In the future, reprogramed regulatory T cells with defined specificity and function may provide life medicines that can persist in patients and achieve lasting disease suppression after one cycle of treatment.

## 1. Introduction

Multiple sclerosis is a chronic neuroinflammatory autoimmune disease in which autoreactive T cells and B cells infiltrate the blood–brain barrier and wrongly attack the myelin sheath, resulting in demyelination and axonal damage in the Central Nervous System (CNS). The condition typically manifests as relapsing–remitting MS (RRMS), although the majority of patients advance to secondary progressive MS (SPMS), which is characterized by disease worsening without signs of remission [1]. About 10–15% of patients with MS are initially diagnosed with primary progressive MS (PPMS), where the disease steadily worsens without any intervals of remission [2,3]. 

Currently, the therapeutic landscape of RRMS revolves around disease-modifying treatments (DMTs) that aim to reduce pathology related to relapses to achieve a favorable clinical course [3]. However, despite a long list of approved DMTs, patients still face the progression of disease and worsening of disabilities. Unfortunately, a DMT’s benefits in patients with RRMS are typically not seen in patients with SPMS and PPMS [2,3]. There is increasing evidence to suggest that the accumulation of axonal loss and neurodegeneration, in association with local inflammation, leads to irreversible neurological disabilities associated with the clinical advancement of MS. Currently, drugs that can promote neural repair are absent [4,5]. Thus, there is a need to develop novel therapies that can provide solutions to the existing unmet needs of patients with MS. 

This review describes how the biological function of regulatory T cells (T_regs_) could be exploited to develop novel cellular therapies for patients with MS. The review aims to introduce the anticipated benefits of T_reg_ therapy to clinical specialists who treat patients with MS, and to engage immunologists with the opportunities and challenges of using genetic engineering to develop an effective life medicine for the targeted treatment of MS.

## 2. CNS Is Not Immune-Privileged

Although the frequency of adaptive immune cells in healthy brain tissue is low, there is a steady state of lymphocyte migration across the blood–brain border, which is often incorrectly described as the blood–brain barrier [5,6,7]. In addition to low-level migration into brain tissue, there is also a population of CNS-resident memory T cells (T_RM_ cells) that, in experimental models, can cause local tissue damage even in the absence of further T cell infiltration from the blood [8,9]. However, the infiltration of pathogenic T cells from the blood into the CNS plays an important role in the progression of MS. Hence, reducing the migration of T cells across the blood–brain border has been a successful strategy in the management of patients with RRMS. The antibody Natalizumab blocks the a4 integrin subunit of VCAM-1 and MDdCAM-1, and thus disrupts T cell adhesion to endothelial cells, resulting in the inhibition of T cell infiltration into the brain and spinal cord [6]. Although Natalizumab and other disease-modifying biologics that prevent T cell infiltration have been effective in RRMS, these interventions are ineffective in SPMS and PPMS. Mechanistically, targeting T cell infiltration can reduce inflammation and restrict cell damage caused by autoreactive immune cells during the relapse–remitting phase of disease, but it cannot reverse the extensive demyelination and tissue damage caused by local T_RM_ cells. Thus, despite the success of treatment with biologics, the current approaches do not enhance tissue repair or stimulate remyelination in patients with MS.

## 3. T_reg_ Biology and Its Role in MS

T_reg_ cells are a subset of CD4+ T cells that are vital to prevent autoimmunity. FoxP3 is the master transcription factor for T_reg_ cells, and inherited defects in the FoxP3 gene lead to the failure of T_reg_ development, which results in fatal autoimmunity in mice [10] and Immune dysregulation, Polyendocrinopathy, Enteropathy, X-linked (IPEX) syndrome in humans [11]. While the majority of T_reg_ cells develop naturally in the thymus, some can be reprogrammed from conventional CD4+ T cells in peripheral tissues. The T cell receptors (TCRs) of thymus-derived T_reg_ cells have been selected for their high affinity to self-peptides [12], and these cells play an important role in maintaining systemic tolerance. By contrast, peripherally induced T_reg_ (pT_reg_) cells are formed in response to low-affinity self-peptides, and in response to commensal microbiota and food antigens, and these cells play a role in tissue-specific hemostasis and tolerance [13]. In vitro culture of conventional CD4+ T cells with TGF-β and IL-2 has been shown to produce induced T_reg_ (iT_reg_) cells that mirror the pT_reg_ lineage commitment pathway [14], yet these iT_reg_ cells are prone to lose FoxP3 expression and thus their T_reg_ functionality. This so-called T_reg_ plasticity is to a large extent under the epigenetic control of the T_reg_-Specific Demethylated Region (TSDR). While the TSDR is completely demethylated in tT_reg_ cells [15], it is partially methylated in pT_reg_ and iT_reg_ cells [16], which may favor the loss of FoxP3 expression in these cells after in vitro restimulation [15,17]. Similarly, activated effector T cells that can temporarily upregulate FoxP3 have a TSDR that is largely methylated [18].

T_reg_ cells are equipped with multifaceted mechanisms to achieve their two major roles—immune suppression and tissue homeostasis (Figure 1). Upon activation by cognate self-antigens and sensing a pro-inflammatory environment, T_reg_ cells are attracted to the site of inflammation, release anti-inflammatory cytokines, and restrict further T cell activation by hindering antigen presentation. Additionally, local immunosuppression can be achieved by activated T_reg_ cells through depriving T cell pro-survival nutrients and signals, such as IL-2, from the local environment [19]. The mechanisms by which T_reg_ cells can repair tissue damage are less well established. Tissue-resident T_reg_ cells can have a direct regenerative ability, as demonstrated in skin and skeletal muscle [20]. Amphiregulin, an EGFR ligand crucial to tissue repair, is dispensable for the suppressive function of T_reg_ cells [21], while its production by T_reg_ cells was found to play a role during the chronic phase after stroke by accelerating neurological recovery [22]. Moreover, the observation that T_reg_ cells can promote oligodendrocyte proliferation and differentiation in vitro may explain how T_reg_ cells can enhance oligodendrocyte-mediated repair of damaged CNS tissue in vivo, as described by Crawford and colleagues [23]. Mechanistic studies further revealed that CCN3, IL-10, and AIM2 expressed by T_reg_ cells play an important role in stimulating neuronal tissue repair [24,25,26,27]. 

Recent studies have uncovered some abnormalities of T_reg_ cells in patients with MS, yet it has been difficult to establish whether these are a cause or consequence of MS pathogenesis. Direct comparisons of the frequency of peripheral T_reg_ cells and infiltrated T_reg_ cells from cerebrospinal fluid between patients with MS and healthy individuals generated inconclusive results [29]. However, ex vivo phenotyping of the peripheral T_reg_ cells of patients with MS revealed reduced FoxP3 expression and impaired suppressive activity [30,31]. A study of patients with untreated RRMS showed an increased number of Th1-like, IFN-γ-producing FoxP3+ T cells compared to healthy controls [32]. This functional plasticity could be due to the highly pro-inflammatory environment of CNS lesions, characterized by IFN-γ, TNF-α, IL-1β, IL-6, and IL-17 [29]. The use of animal models has provided a more mechanistic understanding of the role of T_reg_ cells in MS disease. Using the most widely used MS mouse model, Experimental Autoimmune Encephalomyelitis (EAE), it has been shown that T_reg_ depletion or dysfunction can promote disease progression, which can then be rescued by the adoptive transfer of functional T_reg_ cells [19,29,33,34]. Considering the dual actions of T_reg_ cells, it is likely that the beneficial effect of adoptive T_reg_ therapy is the concerted result of immune-suppressive activities and their ability to stimulate tissue repair. 

## 4. In Vivo Boosting of T_reg_ Cells for MS Treatment

To ultimately achieve an increased number of functional T_reg_ cells in patients with MS, a variety of non-cell-based and cell-based therapies are under development. Whilst the former aim at rescuing the patient’s own T_reg_ repertoire, cell-based therapies replenish the patient with ex vivo-expanded and potentially engineered T_reg_ cells. The toolbox of non-cell-based therapies comprises primarily three synergistic approaches, which are peptides for antigen-specific expansion, variants of IL-2 for preferential T_reg_ survival, and small molecules for reinforcing functionality [35]. Although these strategies have not yet entered the clinic, novel biologic designs aiming at combining all three key approaches of the toolbox are emerging. In particular, a recent proof-of-concept study demonstrated the feasibility and efficacy of microparticles decorated with MHC-II molecules containing peptides of Myelin Oligodendrocyte Glycoprotein (MOG) used as a T_reg_ vaccine, which led to targeted expansion of MOG-specific T_reg_ cells via the incorporation of a modified IL-2 molecule stimulating the IL-2 receptor of T_reg_ cells, as well as simultaneous inhibition of effector T cell proliferation via the release of rapamycin [36]. However, these types of strategies rely on the presence of T_reg_ cells with the vaccine-targeted specificity in patients with MS, and they carry the risk of inadvertently expanding pathogenic effector T cells that have the same specificity. Thus, adoptive transfer of antigen-specific T_reg_ cells represents an alternative therapeutic approach.

## 5. Optimizing Adoptive T_reg_ Cell Therapy for MS

As shown in Figure 2, human T_reg_ cells can be isolated from peripheral blood (PB), umbilical cord blood (UCB), or in vitro differentiated from induced pluripotent stem cells (iPSCs). Autologous T_reg_ cells from PB remain the most common source of T_reg_ cells used in patients [35]. The results of a clinical trial in patients with type 1 diabetes produced safety data of autologous ex vivo-expanded polyclonal T_reg_ cells [37], prompting the development of T_reg_ therapies for other autoimmune diseases, including MS. In addition, several studies with UCB-derived T_reg_ cells have demonstrated that they displayed superior phenotypic stability and greater TCR repertoire diversity compared to PB-derived T_reg_ cells [38,39]. As UCB biobanks expand, the use of an HLA-matched allogeneic T_reg_ population for the treatment of graft-versus-host disease, and certain autoimmune conditions, becomes a realistic possibility [38]. iPSCs are an attractive source for producing large numbers of ‘off-the-shelf’ T_reg_ cells with defined specificity and phenotypic and functional features. However, T_reg_ differentiation from iPSCs is still at an early stage of development, and TCR+FoxP3 transduction combined with Notch-1 ligand stimulation was required to achieve T_reg_ differentiation [40,41]. Additionally, the use of iPSC-derived T_reg_ cells as ‘off-the-shelf’ medicines would still require additional genetic engineering to avoid host-mediated recognition and rejection of the adoptively transferred cells [13]. 

Therapeutic T_reg_ cells can be generated in three ways (Figure 2), which include (i) polyclonal expansion or (ii) antigen-specific expansion of autologous T_reg_ cells obtained from patients, and (iii) genetic engineering of T_reg_ specificity [13]. The observation that multiple autoantigens have been identified as the cause of MS, and more than one clone of autoreactive T cells can be found in patients [42], raised the question whether adoptive transfer of polyclonal T_reg_ cells is preferable over T_reg_ cells with a single specificity. However, T_reg_ cells with specificity for one antigen, for example, Myelin Basic Protein (MBP), can achieve local suppression of effector T cells present in the same microenvironment, even if these effector cells are specific for several other proteins of the neuronal myelin sheath. For example, experiments in the murine EAE model showed that T_reg_ cells specific for MBP were able to suppress pathology caused by effector T cells specific for MOG [43]. The murine EAE model was also used to demonstrate that T_reg_ cells with defined specificity are more potent than polyclonal T_reg_ cells in controlling MS-like symptoms [33,34], which is consistent with better disease control by antigen-specific T_reg_ cells observed in other autoimmune animal models [44,45,46,47,48]. The improved efficacy of antigen-specific over polyclonal T_reg_ cells is related to the enhanced antigen-driven migration of T_reg_ cells into target tissues, and the continued TCR stimulation at the site of pathology, which is required for T_reg_ cells to exert their optimal suppressive function [35]. 

A population of antigen-specific T_reg_ cells can be produced by ex vivo stimulation with antigens, or by genetic T_reg_ engineering. Like conventional T cells, the specificity of natural T_reg_ cells is determined by the TCR that is expressed on the surface of individual cells. The TCR binding to its cognate peptide antigen presented by the Human Lymphocyte Antigen (HLA) class II leads to T_reg_ stimulation and activation of its suppressive function. Similar to conventional T cells, individual T_reg_ cells express distinct TCRs with distinct specificity. Within the natural polyclonal T_reg_ population, the frequency of cells with a TCR of defined specificity, for example, specific for MBP or MOG, is very low, and the expansion of these cells by in vitro stimulation with peptide antigens is technically difficult and unreliable compared to the robust process of genetic engineering of T_reg_ specificity [49]. In the past 10–15 years, specificity engineering of human T cells for adoptive cell therapy has been achieved by the transfer of genes encoding TCRs and CARs (Chimeric Antigen Receptors). Efficient gene transfer protocols, including lentiviral gene transfer, have been developed and result in more than 50% of human T_reg_ cells displaying the antigen specificity that is determined by the introduced TCR or CAR constructs. Whilst antigen recognition by transgenic TCRs is restricted by the HLA genotype of patients, the antigen-binding domain of CARs is a synthetic single-chain variable antibody construct that allows antigen recognition irrespective of the HLA. However, in contrast to TCRs, CARs can only recognize proteins present on the cell surface but not proteins in the cytosol or nucleolus, whereas HLA-presented peptides can be derived from any cellular protein for recognition by TCRs. Hence, the number of cellular proteins that can be targeted by TCRs is vastly greater than the proteins targetable by CARs. Currently, the preclinical development of CAR-T_reg_ cells for MS is lagging behind the use of TCR-engineered T_reg_ cells [50]. 

## 6. The Risks and Challenges of Antigen-Specific T_reg_ Cell Therapy 

The challenges of T_reg_-based adoptive cell therapy include the common limitations of currently available techniques and challenges specific to the treatment of MS. The first challenge is the still inconclusive markers for the isolation of T_reg_ cells with high purity, phenotypic stability, and continued functionality. At present, the markers CD4+ CD25+ CD127-/low are frequently used to purify T_reg_ cells. Unfortunately, staining for FoxP3, an intracellular protein, requires cell permeabilization and is therefore not suitable for the purification of live cells for adoptive therapy, although the vast majority of CD4+ CD25+ CD127-/low cells are FoxP3-positive and thus largely overcome the inability of using FoxP3 for cell purification. However, T_reg_ isolation is further complicated by the observation that FoxP3-positive T_reg_ cells consist of distinct subsets with distinct functionality. For example, it was shown that tT_reg_ cells are more resistant to phenotypic plasticity than pT_reg_ cells [51], yet any differentiating phenotypic markers remain to be elucidated in humans [13]. At present, we do not know which T_reg_ subset is most suitable for dampening the neuroinflammation in the CNS in the context of MS.

## 7. Risks of TCR Gene Therapy

Retro and lentiviral gene transfer, which so far has been used in most human applications of CAR or TCR-T cell therapy, can cause genome toxicity by insertion mutagenesis, which has resulted in fatal side effects with gene-engineered stem cells [52], while similar side effects have not been seen with gene-engineered T cells. Owing to the heterodimer nature of TCR alpha and beta chains, mispairing between transgenic TCRs and endogenous TCRs is a risk associated with TCR but not CAR engineering. Experiments in murine T cell transfer models have shown that TCR mispairing can result in severe side effects in lymphodepleted recipient mice [53]. In patients, toxicities caused by TCR mispairing have not yet been described, although a large number of patients have been treated over the past years. Nevertheless, strategies for modifying transgenic TCRs to avoid pairing with endogenous TCRs, such as cysteine modification [54], domain swapping [55], constant region modification [56], and variable region modification [57], all reduce the risk of mispairing. The CRISPR/Cas9 precision gene-editing tool changed the landscape, as it allows specific disruption of TRAC and TRBC loci, which encode the TCRα and TCRβ chain, respectively. As recently demonstrated, the disruption of endogenous TCR expression and insertion of therapeutic CAR or TCR constructs into the TRAC locus has improved the in vivo functionality of engineered T cell products in preclinical models [58,59]. 

## 8. Improving the Safety of Engineered Treg Therapy

Inevitably, any drugs with immunosuppressive activity have a risk of systemic immunosuppression, whereupon the likelihood of opportunistic infection or malignancy will be elevated. Being a cell therapy, adoptively transferred T_reg_ cells are expected to persist long term and therefore require careful assessment of possible side effects. 

Safety features developed for TCR/CAR engineering for cancer cell therapy can be translated into controlling unwanted systemic immunosuppression of T_reg_-based therapy. One strategy is to use a suicide gene to control the in vivo survival and/or functionality of the adoptively transferred T_reg_ cells. Numerous types of ‘suicide switch’ have been designed and assessed for their efficacy and safety in preclinical studies and in patients, exemplified by truncated EGFR [60] and inducible Caspase 9 [61]. Another approach to regulating T_reg_ function in vivo is the introduction of synthetic receptors. The requirement of IL-2 for T_reg_ proliferation, survival, and function makes IL-2 a promising manipulative pathway [62]. Recently, an elegant study has demonstrated that a genetically modified IL-2 receptor binds selectively to a synthetic IL-2 molecule that contains a ‘matched’ modification, while wild-type IL-2 does not bind to the modified receptor. Hence, engineered T cells expressing the modified IL-2 receptor responded in vivo only to administered synthetic IL-2 containing the ‘matched’ modification but not to endogenous IL-2 [63]. This technology could be used to engineer T_reg_ cells whose survival and function in vivo is controlled by the time and dose of synthetic IL-2 administration.

The functional plasticity of T_reg_ cells has been demonstrated repeatedly both in vitro and in vivo following adoptive transfer. Upon cues from inflammatory cytokines and IL-2 deficiency, antigen-specific T_reg_ cells can adopt an alternative transcriptional program and become effector T cells that may contribute to tissue pathology [64]. For example, in the murine EAE model researchers identified a population of exFoxP3-T_reg_ cells that had lost expression of FoxP3, which was associated with the production of the effector cytokine IFN-g [65]. However, genetic engineering can be used to reinforce the T_reg_ identity by forcing FoxP3 expression. This can be achieved by delivering a genetic construct that drives FoxP3 expression from a constitutively active promotor [43], or by inserting a constitutively active enhancer/promotor into the genome proximal to the FoxP3 gene [66]. Both approaches lead to sustained levels of FoxP3 expression, which ‘locks’ T_reg_ cells into a stable phenotype. Both approaches can also be utilized to force FoxP3 expression in conventional CD4+ T cells and convert them into T_reg_-like cells that display suppressive activity in vitro and in vivo [67]. 

## 9. Conclusions

With their multifaceted immunosuppressive functions and remyelinating capacity, T_reg_ cells are undoubtedly an ideal cellular candidate for MS immunotherapy. Adoptive therapy with engineered T_reg_ cells is now at the exciting stage where the first human clinical trials are ongoing, and will soon provide eagerly awaited feasibility, safety, and efficacy data. Success of T_reg_ therapy will depend on the selection of suitable target antigens that stimulate the suppressive activity and the tissue repair function of T_reg_ cells at the local site of disease, without any systemic impairment of the immune system.

## 10. Future Directions

Despite extensive preclinical research supporting the feasibility and efficacy of engineered Tregs expressing CARs or TCRs, there are currently no results of clinical trials in patients. This will change in the next few years when ongoing clinical trials will report their results. The manufacturing of sufficient numbers of clinical-grade Tregs for the treatment of large numbers of patients remains a substantial challenge in the field. Unlike the production of CAR/TCR-engineered conventional T cells, the frequency of Tregs is much lower, and they require distinct in vitro conditions to achieve successful gene transfer and expansion of the engineered Tregs to produce the therapeutic numbers required for the treatment of patients. The conversion of conventional CD4 T cells, by FoxP3 plus CAR/TCR gene transfer, provides a strategy to substantially increase the cell numbers available for the production of a therapeutic product. In the future, it may be possible to move away from producing personalized T_reg_ products for each patient and use instead the same cell product to treat a large number of patients. The most promising strategy to achieve this is the production of a T_reg_ cell bank from induced pluripotent stem cells, which can provide a ‘permanent’ source of cells with defined biological and functional characteristics. Extensive gene editing of this ‘permanent’ cell source will be required to remove proteins that are the target of rejection and add proteins that protect the cells from immune-mediated attack. A long-term goal is to achieve the in vivo production of engineered Tregs. The production of virus-like particles or biochemically assembled nanoparticles that selectively bind to proteins expressed by Tregs provides a strategy to achieve T_reg_ engineering in patients. The same therapeutic product, a preparation of virus-like vectors or nanoparticles, would be injected into many patients to reprogram Tregs in vivo. Although cell-specific in vivo gene engineering already exists, more work is needed to assess the efficacy and the safety profile of this exciting approach.

## Figures and Tables

**Figure 1 cells-13-00797-f001:**
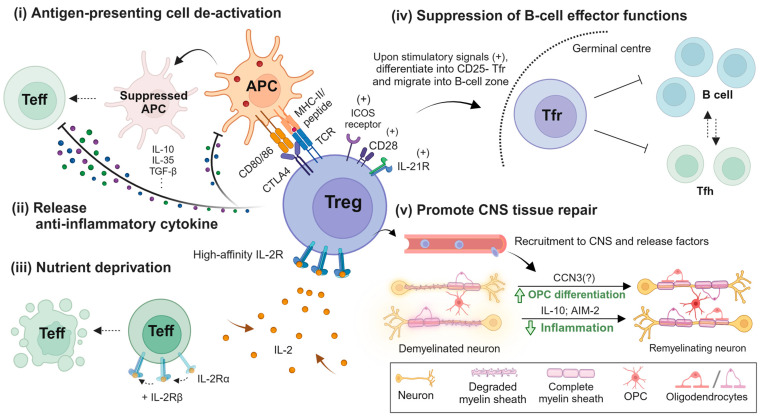
**The beneficial immunosuppressive and tissue repair mechanisms of actions of T_reg_ cells in multiple sclerosis.** In lymphoid organs, resting T_reg_ cells will be activated by the cognate antigen presented by Antigen-Presenting Cells (APCs) and elicit immunosuppressive functions, including APC suppression through (i) dampening the costimulatory molecules CD80/86 by CTLA4 binding; (ii) the production of anti-inflammatory cytokines, such as IL-10 and TGF-β; and (iii) scavenging signals and metabolites essential to the survival and functionality of pro-inflammatory cells, as well as suppressing B cell effector functions for high-affinity antibody production [28]. On the other hand, once attracted to the inflammatory CNS, T_reg_ cells can produce IL-10 and AIM2, which dampens the neuroinflammation, and CCN3, which potentiates oligodendrocyte differentiation for remyelination. The figure was produced by Biorender.com.

**Figure 2 cells-13-00797-f002:**
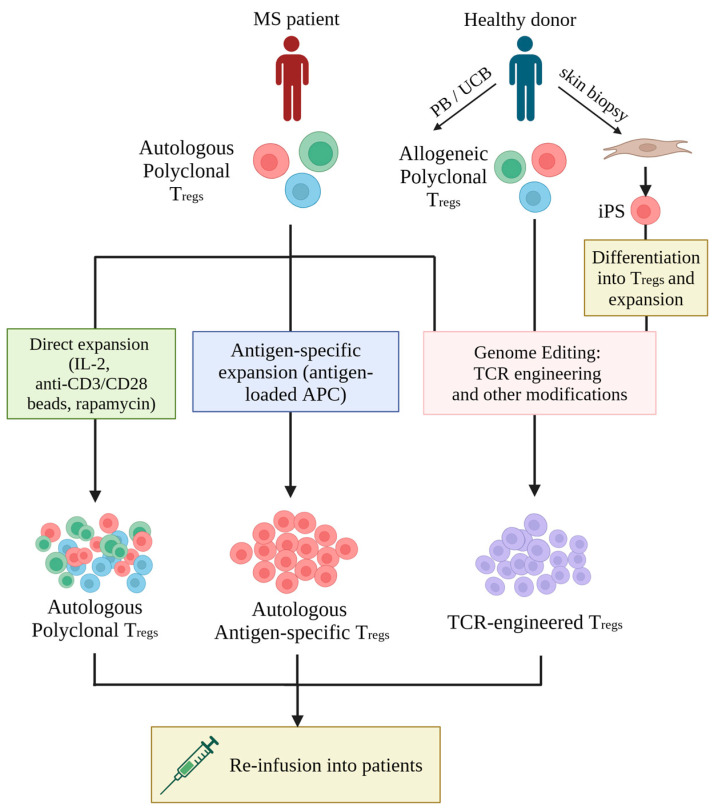
**Summary of T_reg_-based therapy.** Several promising sources, including the T_reg_ cells extracted from the peripheral blood (PB) or umbilical cord blood (UCB) or developed from the fibroblast-to-induced pluripotent stem (iPS) cell developmental pathway. The figure was produced by Biorender.com.

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
