# Peer review of "Targeted Therapy of Multiple Sclerosis: A Case for Antigen-Specific Tregs"

_cells, 2024, doi:10.3390/cells13100797_

Round 1

Reviewer 1 Report

Comments and Suggestions for Authors

In this review, the authors reviewed the advancement of developing targeted novel therapy in multiple sclerosis with a focus on antigen-specific Treg-based therapeutics. Despite its comprehensive content, the paper suffers from a lack of clear logical flow and adequate explanations for the intended readership. The review paper needs major revision before the acceptance.

1.       The introduction should conclude with a discussion on the primary focus of this review to guide readers.

2.     There are many unexplained abbreviations; these should be explained upon their first occurrence. For instance, clarify the meaning of 'iTreg' at Line 79.

3.       In part 6, detailed information should be provided about TCR and CAR Treg therapies, especially their role in targeting multiple sclerosis. More clarification is needed on whether TCR and CAR Treg methods fall under the category of antigen-specific Treg therapies.

4.       It is recommended to put the part 7 right after part 3 to enhance the coherence of the manuscript.

5.       It is also recommended to merge part 4, part 5 and part 6 to a single section focused on reviewing the advancements in novel Treg therapies and their methodologies respectively.

6.       Please have more specific subtitles for part 9-11. For example, in part 9, the authors focused on discussing genome toxicities and mispairing toxicities. It can be more appropriate to have “Toxic challenges of TCR-based therapy” instead of generic “Off target toxicities” for this part.

7.       There are many grammatical and punctuation mistakes. For instance, Change “formed in response low…” to “formed in response to low”.  Change “self-peptides and …” to “self-peptides, and”.

Comments on the Quality of English Language

Moderate editing of English language required

Author Response

In this review, the authors reviewed the advancement of developing targeted novel therapy in multiple sclerosis with a focus on antigen-specific Treg-based therapeutics. Despite its comprehensive content, the paper suffers from a lack of clear logical flow and adequate explanations for the intended readership. The review paper needs major revision before the acceptance.

  1. The introduction should conclude with a discussion on the primary focus of this review to guide readers.

Reply: We have added a paragraph at the end of the introduction to explain the focus of this review.

  1. There are many unexplained abbreviations; these should be explained upon their first occurrence. For instance, clarify the meaning of 'iTreg' at Line 79.

Reply: We apologise that unexplained abbreviations have appeared in the text.  We checked the text and explained abbreviations where this was not the case.

  1. In part 6, detailed information should be provided about TCR and CAR Treg therapies, especially their role in targeting multiple sclerosis. More clarification is needed on whether TCR and CAR Treg methods fall under the category of antigen-specific Treg therapies.

Reply: We have substantially modified part 6 to address the concerns of the reviewer

  1. It is recommended to put the part 7 right after part 3 to enhance the coherence of the manuscript.

Reply: We thank the reviewer for this suggestion.  We have moved part 7 now after part 4, to mark here the start of strategies for adoptive Treg therapy.  Part 4 covers strategies to manipulate Tregs in vivo without adoptive cell transfer.  We hope that the reviewer agrees with this rationale.

  1. It is also recommended to merge part 4, part 5 and part 6 to a single section focused on reviewing the advancements in novel Treg therapies and their methodologies respectively.

Reply: Following the rationale above, we have merged the previous part 7 with part 5 and 6 to create a new part entitled “Optimizing adoptive Treg cell therapy for MS”

  1. Please have more specific subtitles for part 9-11. For example, in part 9, the authors focused on discussing genome toxicities and mispairing toxicities. It can be more appropriate to have “Toxic challenges of TCR-based therapy” instead of generic “Off target toxicities” for this part.

Reply: We have renamed part 9 as ‘Risks of TCR gene therapy’ and merged part 10+11 under the heading ‘Improving the safety of engineered Treg therapy’.  We hope that this addresses the reviews request for more specific subtitles.

  1. There are many grammatical and punctuation mistakes. For instance, Change “formed in response low…” to “formed in response to low”.  Change “self-peptides and …” to “self-peptides, and”.

Reply: We apologies for these oversights and have corrected these mistakes

Reviewer 2 Report

Comments and Suggestions for Authors

Zhong and Stauss wrote a timely and interesting review on an emerging field: potentials of multiple sclerosis therapy with regulatory T-cells. Prof. Stauss is a well established scientist in the field with multiple commercial enterprises running on the Treg field. Here is my first point:

1.) I could not find a conflict of interest declaration.

2.) While the manuscript is well written in fluent english some times the format is changed to italic such a line 79 to 86!

3.) Some specific points might need further attention: Crucial for the concept of therapeutic Tregs specifc for MBP is the presence of such cells within the leasoned brain. Here references are scarce with only one old reference Engelhardt and Ransohoff 2012). Please, elaborate this important point with new references and the concept CNS-resident T-cells.

4.) Figure 1 seems to be recycled from the topic of tissue repair which Quell pharamceuticals used for liver. Neither muscle injury nor skin injury are the topic of this review. Please, delete these parts and concentrate on the effects of Tregs in MS. Which factors drive OPC differentiation? Which factors control inflammation of neurons and oligodendrocytes? Please, inrease size of the right side of Fig. 1. so that it is readable.

5.) Furthermore on the left side the effects of Tregs on B-cells are missing, please elaborate by discussing e.g. Sage, P. T. et al. Suppression by TFR cells leads to durable and selective inhibition of B cell effector function.(Nat. Immunol. http://dx.doi.org/10.1038/ni.3578 (2016)).

6.) Fig. 2 is well designed, Risks and Challenges are interesting and the conclusion is clear.

Author Response

Zhong and Stauss wrote a timely and interesting review on an emerging field: potentials of multiple sclerosis therapy with regulatory T-cells. Prof. Stauss is a well established scientist in the field with multiple commercial enterprises running on the Treg field. Here is my first point:

  • I could not find a conflict of interest declaration.

Reply: Apologies for this oversight.  The conflicts have now been added to the paper.

  • While the manuscript is well written in fluent english some times the format is changed to italic such a line 79 to 86!

Reply: The italic section seems a consequence of the processing of the submitted paper.  The submitted word document did not have this section in italic.  We apologise for this.

  • Some specific points might need further attention: Crucial for the concept of therapeutic Tregs specifc for MBP is the presence of such cells within the leasoned brain. Here references are scarce with only one old reference Engelhardt and Ransohoff 2012). Please, elaborate this important point with new references and the concept CNS-resident T-cells.

Reply: . We apologise that our references were incomplete. We have inserted additional references with particular focus on the role of CNS resident T cells in MS

  • Figure 1 seems to be recycled from the topic of tissue repair which Quell pharamceuticals used for liver. Neither muscle injury nor skin injury are the topic of this review. Please, delete these parts and concentrate on the effects of Tregs in MS. Which factors drive OPC differentiation? Which factors control inflammation of neurons and oligodendrocytes? Please, inrease size of the right side of Fig. 1. so that it is readable.

Reply: We did not use any original or modified material from Quell Therapeutics for this review.  However, we have followed the reviewer’s suggestion to modify the figure as outlined by the reviewer.

  • Furthermore on the left side the effects of Tregs on B-cells are missing, please elaborate by discussing e.g. Sage, P. T. et al. Suppression by TFR cells leads to durable and selective inhibition of B cell effector function.(Nat. Immunol. http://dx.doi.org/10.1038/ni.3578 (2016)).

Reply: We have added the Treg effect on B-cells as suggested by the reviewer

  • 2 is well designed, Risks and Challenges are interesting and the conclusion is clear.

Reply: Thank you

Round 2

Reviewer 1 Report

Comments and Suggestions for Authors

The revised manuscript needs minor revision and then will be good to be accepted.

There are two more abbreviations that needs to be clarified as follows: 

1. TRM in line 58

2. pTreg in line 87

Author Response

We have clarified the two abbreviations: 

1. TRM in line 58

2. pTreg in line 87